# Parents' perspectives on a national child oral health promotion program: Sociodemographic influences and behavioral insights – A cross-sectional analysis

**Mohammad Reza Khami[1,2], Mohammadreza Naderi[3], Shabnam Varmazyari[1,4]***

**1** Research Center for Caries Prevention, Dentistry Research Institute, Tehran University of Medical Sciences, Tehran, Iran, **2** Department of Community Oral Health, School of Dentistry, Tehran University of Medical Sciences, Tehran, Iran, **3** Department of Oral and Maxillofacial Surgery, TeMS.c., Islamic Azad University, Tehran, Iran, **4** Laser Research Center of Dentistry, Dentistry Research Institute, Tehran University of Medical Sciences, Tehran, Iran

* Shabnam.varmazyari@gmail.com

## Abstract

### Background

Iran's *Students' Oral Health Promotion Program* (SOHPP) aimed to improve primary school children's oral health, but parental perceptions of this program, as key stakeholders, remain underexplored. This study explores parents' perceptions of Iran's SOHPP, the sociodemographic factors shaping them, and children's post-program oral health behaviors.

### Methods

Conducted at four randomly-selected comprehensive healthcare centers in Tehran (July–August 2020), this cross-sectional study phone-surveyed parents of primary school children who participated in Iran's SOHPP. The questionnaire covered socio-demographics, children's post-program oral health behaviors, and awareness and satisfaction with key SOHPP components: oral health education, fluoride therapy, electronic oral health profiling, and treatment need identification. ANOVA, chi-square, and regression models served for statistical analysis.

### Results

The 354 surveyed parents (response rate: 67%), on average, scored 79% for SOHPP awareness and 74% for satisfaction. Awareness and satisfaction were lowest for treatment-related components (58.2% and 52.0% for oral health profiling; 70.9% and 53.7% for treatment need identification). Fluoride therapy acceptance was 76.6%, with refusals mainly due to poor notification and limited procedural understanding. While 61.6% of parents noted improved tooth-brushing in their child, post-program,

**Data availability statement:** All relevant data, especially, the minimal data set supporting this article, are provider within the manuscript itself and the uploaded Supporting Information files.

**Funding:** This study was supported in part by funding from Tehran University of Medical Sciences as a student thesis grant.

**Competing interests:** The authors have declared that no competing interests exist.

only 38.7% reported twice-daily brushing, and 37.9% were unaware of their fluoride toothpaste use. Additionally, 41.5% reported sugary snacking at least once daily by their child, while 83.0% reported healthy school food intake. More educated fathers had greater program awareness (B = 0.18, p = 0.040), satisfaction (B = 0.17, p = 0.032), and fluoride therapy acceptance (OR = 1.37, p = 0.024), whereas government-employed household heads were less aware (B = −1.16, p = 0.004) and less likely to perceive tooth-brushing improvements (B = −1.48, p = 0.001).

## Conclusions

To improve parents' perceptions of Iran's SOHPP, enhanced delivery of treatment-related components, improved fluoride therapy transparency, reinforced post-program oral health behaviors, and tailored outreach to fathers' education and household heads' government employment are recommended.

## Introduction

Childhood oral health remains a major global public health concern, with 60–90% of school-aged children affected by dental caries worldwide [1,2]. These conditions substantially impair children's daily functioning, learning capacity, and social development [3–5], with the burden falling disproportionately on low- and middle-income countries (LMICs) such as Iran [6,7]. In Iran, childhood oral health challenges are marked by high levels of untreated caries, low proportions of caries-free children, and progressive deterioration of permanent teeth and periodontal health with age [8,9]. These problems are further exacerbated by weak regulatory frameworks, resource constraints, and inequitable access to services [9–13].

In response to the childhood oral health crisis, child oral health promotion programs gained prominence for improving clinical outcomes, strengthening knowledge and behaviors, expanding access to prevention, and lowering system costs [14–18]. In Iran, the Students' Oral Health Promotion Program (SOHPP), launched in 2015, represents a nationwide initiative targeting the country's nine million primary school children. Throughout the program, dentists trained teachers at schools and health-care providers at comprehensive health centers, Iran's primary healthcare hubs [9], to in turn, deliver oral health education to children and parents. Preventive care was provided as biannual fluoride varnish in schools, and treatment involved school-based screenings, establishment of electronic oral health records, and referrals to public and private care providers [9,19].

Meaningful stakeholder engagement is a cornerstone of effective community-based oral health programs, with routine assessment of stakeholder views informing program effectiveness, continuity, and accountability [20–23]. Parents are central stakeholders in child oral health, as they shape knowledge, attitudes, and behaviors, and influence long-term clinical outcomes [20,24–27]. Moreover, parental perspectives and engagement are central to the success of child oral health promotion programs, as they are associated with improved clinical outcomes, reinforce efforts at

home, strengthen program fidelity, quality, and long-term sustainability [28–32], and are endorsed by both the Centers for Disease Control and Prevention and the World Health Organization [32–34].

Despite this, parental engagement and perceptions of child oral health promotion programs remain largely underexamined. Most international studies have emphasized clinical and behavioral outcomes, assessing parental perspectives inconsistently and predominantly in small, context-specific studies from high-income settings [20,35,36]. Regionally, evidence is similarly limited; few studies such as the evaluation of Qatar's *Asnani* program have examined parental perspectives, largely confined to overall satisfaction and perceived behavioral change [37]. In Iran, existing research has focused mainly on children's post-SOHPP caries outcomes [23,38], and one study uniquely evaluated children's experiences of the program [39]. Moreover, prior studies that did examine parental engagement and perceptions paid little attention to perceptions of specific program components or their sociodemographic correlates [20,35–37].

Therefore, the present study aims to provide overall and component-level assessments of parents' perceptions of Iran's SOHPP, examine sociodemographic correlates of these perceptions, and situate them alongside parent-reported post-program child oral health behaviors. For the inferential objectives, the null hypotheses are that sociodemographic characteristics are not associated with any parent perspectives. By generating evidence on this issue, this study hopes to provides a basis for refining the SOHPP's design, tailoring its implementation to diverse families, and promoting its equitable uptake.

## Methods

### Study design and ethical considerations

This is a quantitative cross-sectional analytical study based on a structured telephone survey; no qualitative or mixed-methods data collection or analysis was performed. The study was conducted between July 1st and August 31st 2020, after obtaining approval from the Ethics Committee of Tehran University of Medical Sciences (TUMS) (ethics code: IR.TUMS.DENTISTRY.REC.1399.011). All study procedures adhered to the principles of the Declaration of Helsinki. Participation was voluntary, and verbal informed consent was obtained at the beginning of each telephone interview, as the survey format rendered written consent impractical. Prior to consent, interviewers clearly explained the voluntary nature of participation, the right to withdraw at any time, and the measures taken to ensure confidentiality and anonymity. These measures included obtaining phone numbers from the Health and Treatment Deputies of TUMS without any accompanying personally identifiable information and administering all questionnaires anonymously. Verbal consent was documented by the interviewer in a secure log, recording the date, time, participant's anonymized identification code, and confirmation of consent. The TUMS Ethics Committee reviewed and approved this consent procedure as appropriate for the study.

### Sample Size calculation

Since no sufficiently similar prior research was found, the main sample size was estimated using a pilot study on 30 parents. Based on the pilot study, following the standard sample size calculation pathway for quantitative proportion-based surveys, and using the one-sample proportion confidence interval function in SPSS (IBM SPSS Statistics; version 26, IBM Corp., Armonk, NY, USA) the minimum required sample size was calculated as 347, assuming $\alpha = 0.05$, $p = 0.69$, and $d = 0.1$.

### Study setting, population, and procedure

The study was conducted at four randomly selected Comprehensive Healthcare Centers within TUMS jurisdiction in Tehran city, Iran. Iran's healthcare system is overseen by its medical universities, and in Tehran, the southern region falls under TUMS's jurisdiction [40]. Tehran city was chosen for its logistical feasibility, large population, and status as the nation's capital, while the TUMS jurisdiction was selected for its high population density and accessibility [41,42].



Parents were eligible to participate if they had a child in grades 1–6 that had participated in the SOHPP, were registered at one of the selected Comprehensive Healthcare Centers under TUMS jurisdiction in Tehran city, had a reachable phone number on record, and provided informed consent.

Data for this study were collected using a structured questionnaire. The research team first obtained the necessary approvals from TUMS and requested a list of comprehensive healthcare centers under their supervision. The full list of these centers was entered into an Excel spreadsheet (Microsoft Excel; Microsoft Corp., Redmond, WA, USA). A random number was generated for each center using the RAND() function in Excel and the list was sorted by these values. The top four centers, Ayat, Farmafarmanian, Imam Hassan Mojtaba, and Avicenna, were then selected. The research team visited these centers to obtain parent contact information. Then, one of the researchers conducted telephone interviews with parents using a questionnaire developed for the study.

## Study tool and measures

Data were collected using a Persian questionnaire developed for this study (supporting file 1). This questionnaire was checklist-based in nature, did not evaluate complex constructs (such as oral health knowledge or attitudes) across multiple domains or with Likert scales, and merely captured whether parents were aware of, satisfied with, and accepting of specific components of the SOHPP. In addition, only one member of the research team administered this questionnaire via telephone interviews, who was actively available to explain each item, clarify response options, and ensure accurate understanding throughout the call. Therefore, simplified psychometric validation procedures were deemed appropriate for it by the Community Oral Health faculty of TUMS. Face and content validity were assessed by group discussions among five faculty experts in Community Oral Health, who reviewed the items for relevance, clarity, and comprehensiveness over iterative rounds. Feedback from these procedures was used to revise the questionnaire where necessary, resulting in minor revisions to item wording and instructions. Reliability was evaluated using a test-retest method with a subsample of 20 parents, with the retest administered two weeks after the initial test. The test-retest approach focused on identifying inconsistencies for qualitative revisions, rather than formal statistical metrics. Review of responses showed high consistency across most items (approximately 85% agreement across items), with revisions made to improve clarity.

The instrument included four main sections that asked about: 1) parents' sociodemographic characteristics, 2) their awareness and satisfaction with the four key SOHPP components (oral health education, fluoride therapy, electronic oral health profiles, and treatment need identification), 3) their acceptance/perceived impact of certain program components, and 4) their reports on post-program oral health behaviors of their child. More details on the items in these sections, their response options, and how they were grouped to form the research measures are provided below:

Sociodemographic characteristics: This section included items on the respondent's relation to the child (mother/father/other), father's level of education (illiterate, literate, primary school, middle school, high school diploma, associate degree, bachelor's degree, master's degree, doctorate or higher, do not know), mother's level of education (similar items); household size (in discrete numbers), household head occupation (government employee, government sector worker, private sector employee, private sector worker, self-employed, homemaker, retired, unemployed), and the child's school grade (numbers between 1–6). These characteristics were treated as independent variables in analyses.

The derived outcome variables included:

a) Overall awareness of SOHPP: Defined as having awareness that SOHPP exists and offers specific services, did not imply approval or participation. This factor was assessed using four items covering oral health education, fluoride therapy, electronic oral health profiling, and identification of treatment needs. Responses were coded as Yes = 2, No opinion = 1, No = 0. A composite awareness score ranging from 0–8 was calculated by summing scores for these 4 items, with higher scores indicating greater awareness.

b) Overall satisfaction with SOHPP: Defined as perceived service quality and subjective evaluation of program delivery, not behavioral compliance or endorsement. This factor was assessed using four items corresponding to the same above noted four program components. Responses followed the same coding scheme as the awareness variable. A composite satisfaction score ranging from 0–8 was calculated by summing scores for these 4 items, with higher scores reflecting greater satisfaction

c) Acceptance of fluoride therapy: Defined as behavioral consent to the intervention. This factor was assessed through a single item asking whether parents agreed with fluoride application for their children. Response options included "Yes", "No opinion", and "Other", with the parents stating their reason for refusal if opting for the last option.

d) Perceived impact of SOHPP on child's tooth-brushing habits: Defined as parent's belief that the SOHPP had influenced their child's tooth-brushing behavior, reflecting perceived behavioral change, not objective behavior. This factor was assessed through a single item asking whether the parent thought the program had influenced their child's tooth-brushing behavior. Response options included "Positive", "No opinion", and "Negative".

e) Daily tooth-brushing frequency: This factor was assessed through a single item asking how often the child brushed their teeth per day post-program implementation. Response options included "At least twice a day", "Once a day", "Less than once a day", and "Never".

f) Use of fluoride toothpaste: This factor was assessed through a single item asking if the child used fluoride toothpaste during tooth-brushing post-program implementation. Response options included "Yes", "No", and "No opinion".

g) Frequency of sugary snack consumption: This factor was assessed through a single item asking how frequently the child consumed sugary snacks post-program implementation. Response options included "At least once a day", "Less than once a day", and "Never".

h) Type of foods consumed at school: This factor was assessed through a single open-ended item asking what types of foods the child typically ate at school post-program implementation. One of the researchers reviewed the responses and classified them according to the World Health Organization's (WHO) nutritional criteria [40], which defines healthy foods as those low in added sugars, saturated fats, and sodium and unhealthy foods as those high in added sugars, saturated fats, or sodium. For example, responses listing "milk," "bread with cheese," or "fruits" were coded as "Healthy," while "chips," "chocolate," or "sugar-sweetened drinks" were coded as "Unhealthy."

## Statistical analysis

Data were analyzed using SPSS (IBM SPSS Statistics; version 26, IBM Corp., Armonk, NY, USA. Descriptive statistics including means, standard deviations, frequencies, and percentages were computed to summarize demographic characteristics and key variables related to parental awareness, satisfaction, and behavioral outcomes. Group comparisons were conducted using ANOVA, Chi-square tests, and independent sample t-tests where appropriate.

To examine associations between independent and outcome variables, multiple linear regression analyses were conducted for continuous outcomes (overall awareness, overall satisfaction, and perceived impact on tooth-brushing), while binary logistic regression was used for the dichotomous outcome (acceptance of fluoride therapy). For both linear and binary regression models, the Enter method was used, which is the standard, go-to model for exploratory, hypothesis-generating studies without strong a priori justification. Model fit, assumptions of normality, linearity, and homoscedasticity were all evaluated. Multicollinearity was assessed using variance inflation factors (VIFs), all of which were below 5, indicating acceptable levels.

Occupation, a nominal variable, was dichotomized for inclusion in the regression models. Categories with small sample sizes, such as unemployed and retired individuals, had to be excluded. Government employees were coded as 0, while



all other occupations were coded as 1, reflecting the considerable differences observed in scores between these groups. Statistical significance was set at $p < 0.05$ for all tests and missing data were handled using listwise deletion.

## Results

To reach the minimum required sample size while accounting for anticipated non-response, a total of 525 individuals were approached, ultimately resulting in 354 participants (a 67% response rate). An average of 87 participants was recruited per comprehensive healthcare center. Fathers comprised the largest group of respondents (47.2%). Among fathers, those with a college degree or higher were most common (30.9%), and those with primary education or less were least common (12.9%). For mothers, high school graduates predominated (28.2%), with primary education or less being the least common (19.2%). Regarding household head occupation, nearly half of the sample were self-employed (47.5%), while retirees were the smallest group (1.7%). About two-fifths of the households had four members (42.9%), and those with seven or more members were rare (2%). Overall, parents of children in grades 1–4 represented nearly 90% of the sample (Table 1).

The mean parental awareness score was 6.33 (SD = 1.97), corresponding to 79% of the maximum score. Overall, 71.6% of parents scored 8, indicating full knowledge of SOHPP implementation, while 13.4% scored 0, reflecting complete unawareness. Awareness varied by program component, from 80.8% for oral health education to 58.2% for electronic oral health profile establishment (Table 2). The mean satisfaction score was 5.90 (SD = 1.72), or 74% of the maximum, with 58.8% of parents scoring 8 (complete satisfaction) and 11.2% scoring 0 (complete dissatisfaction). Satisfaction also varied by program component, from 68.6% for oral health education to 52% for electronic profile establishment. Additionally, 61.6% of parents perceived a positive impact of SOHPP on their child's tooth-brushing habits, while 76.6% accepted fluoride therapy and 15.8% declined. Parents' reasons for refusing school-based fluoride therapy are ranked below by frequency of citation:

1. Lack of prior notification from schools about the intervention

2. Limited knowledge about fluoride therapy procedure

3. Concerns of improper provision in school settings

4. Belief that fluoride therapy increases susceptibility to dental caries

5. Fear of side effects such as gum or tongue redness and burning

6. Preference for a dental office

7. Child-specific issues such as psychological or physical conditions

8. Uncertainty regarding whether or not consent was required for the procedure

As demonstrated in Table 3, approximately two-fifths of parents (38.7%) reported that their child brushed at least twice daily after the SOHPP. Nearly three-fifths (58.8%) said their child used fluoride toothpaste, although about two-fifths (37.9%) were unsure of its use. Meanwhile, 41.5% indicated that their child consumed sugary snacks at least once a day, whereas a large majority (83%) reported that their child ate healthy foods at school. The healthy items mentioned were milk, bread with cheese, fruits and vegetables, and nuts, while unhealthy items were dried sour plums, chips, puffed snacks, chocolate, chewing gum, and sugar-sweetened beverages.

One-way ANOVA indicated significant differences in mean overall satisfaction scores across levels of fathers' education ($p = 0.011$). Post-hoc comparisons revealed that fathers with primary education or lower reported lower satisfaction compared to those with high school diplomas or associate degrees or higher. Similarly, there were significant variations in overall awareness scores by fathers' education ($p < 0.001$), with post-hoc analyses indicating progressively higher

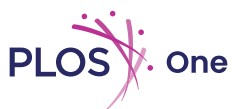

**Table 1. Sociodemographic characteristics of parents of Tehran's primary school children participating in the SOHPP[a] (n = 354).**

| Number (%) | Category | Variables |
|---|---|---|
| 46 (12.9%) | Primary education or lower | Father's education level |
| 101 (28.5%) | Middle school education | |
| 98 (27.7%) | High school diploma | |
| 109 (30.9%) | Associate degree or higher | |
| 68 (19.2%) | Primary education or lower | Mother's education level |
| 91 (25.7%) | Middle school education | |
| 100 (28.2%) | High school diploma | |
| 95 (26.9%) | Associate degree or higher | |
| 168 (47.5%) | Self-employed | Household head's occupation |
| 60 (16.9%) | Private-sector laborer | |
| 52 (14.7%) | Private-sector employee | |
| 31 (8.8%) | Government laborer | |
| 25 (7.1%) | Government employee | |
| 12 (3.4%) | Unemployed | |
| 6 (1.7%) | Retiree | |
| 90 (25.4%) | 3 | Household size |
| 152 (42.9%) | 4 | |
| 64 (18.1%) | 5 | |
| 41 (11.6%) | 6 | |
| 7 (2%) | 7 or higher | |
| 146 (41.3%) | Mother | Relation of respondent to the child |
| 169 (47.7%) | Father | |
| 39 (11%) | Other | |
| 82 (23.2%) | First | Child's school grade |
| 83 (23.4%) | Second | |
| 87 (24.6%) | Third | |
| 69 (19.5%) | Fourth | |
| 28 (7.9%) | Fifth | |
| 5 (1.4%) | Sixth | |

[a]SOHPP: Students' oral health promotion program.

awareness among those with middle school education, high school diplomas and associate degrees or higher. For mothers' education, only awareness scores differed (p = 0.018), driven by higher means among those with high school diplomas compared to primary education or lower. Household head occupation was associated with significant differences in awareness (p = 0.007), with unemployed heads reporting the lowest scores (mean = 4.58, SD = 2.2), though satisfaction did not vary significantly (p = 0.401) (Table 4).

As shown in Table 5, higher father education was associated with greater overall awareness of the program (B = 0.18, p = 0.040), higher overall satisfaction (B = 0.17, p = 0.032), and greater approval of the child receiving fluoride therapy (OR = 1.37, p = 0.024). In contrast, being a government employee as a household head was related to lower overall awareness of the program (B = −1.16, p = 0.004) and less likelihood of deeming the program had had a positive impact on child tooth-brushing frequency (B = −1.48, p = 0.001).



**Table 2. Perceptions of SOHPP[a] components among parents of Tehran's primary school children (n = 354).**

| Variable | Category | N (%) |
|---|---|---|
| Awareness of electronic oral health profile establishment | No | 72 (20.3%) |
| | No opinion | 76 (21.5%) |
| | Yes | 206 (58.2%) |
| Awareness of fluoride therapy provision | No | 30 (8.5%) |
| | No opinion | 53 (15.0%) |
| | Yes | 271 (76.6%) |
| Awareness of oral health education | No | 29 (8.2%) |
| | No opinion | 39 (11.0%) |
| | Yes | 286 (80.8%) |
| Awareness of treatment need identification | No | 59 (16.7%) |
| | No opinion | 44 (12.4%) |
| | Yes | 251 (70.9%) |
| Satisfaction with electronic oral health profile establishment | No | 15 (4.2%) |
| | No opinion | 155 (43.8%) |
| | Yes | 184 (52.0%) |
| Satisfaction with fluoride therapy provision | No | 59 (16.7%) |
| | No opinion | 79 (22.3%) |
| | Yes | 216 (61.0%) |
| Satisfaction with oral health education | No | 43 (12.1%) |
| | No opinion | 68 (19.2%) |
| | Yes | 243 (68.6%) |
| Satisfaction with treatment need identification | No | 42 (11.9%) |
| | No opinion | 122 (34.5%) |
| | Yes | 190 (53.7%) |
| Fluoride therapy acceptance | Yes | 271 (76.6%) |
| | No | 56 (15.8%) |
| | Other | 27 (7.6%) |
| Perceived Impact on tooth-brushing habit | Negative | 84 (23.7%) |
| | No opinion | 52 (14.7%) |
| | Positive | 218 (61.6%) |

[a]SOHPP: Students' oral health promotion program.

## Discussion

Iran's SOHPP offers a noteworthy example of large-scale, national child oral health promotion program, integrating oral health education, prevention, and treatment into routine student health and leveraging intersectoral collaboration to reach diverse family dynamics. This study examined parents' perceptions of Iran's SOHPP, highlighting positive trends and remaining gaps while stressing the need for tailored strategies to improve parental views. Parents showed generally high awareness and satisfaction with the SOHPP, though both were lowest for the treatment-related components. Acceptance of fluoride therapy was also high, with refusals largely tied to lack of prior notification and limited understanding of the procedure. A notable perception–behavior gap emerged in tooth-brushing habits, as 61.6% of parents perceived improved brushing post-program, yet only 38.7% reported twice-daily brushing. Moreover, 37.9% of parents were unaware whether their child used fluoride toothpaste, and their reports on the children's dietary behaviors were mixed. Finally, the findings

**Table 3. Parental reports of the oral health behaviors of Tehran's primary school children post-SOHPP[a] (n = 354).**

| Variable | Category | Number (%) |
| --- | --- | --- |
| Tooth-brushing frequency | Twice or more per day | 137 (38.7%) |
| | Once per day | 125 (35.3%) |
| | Less than once per day (irregular) | 69 (19.5%) |
| | Never | 23 (6.5%) |
| Use of fluoride toothpaste | Uses fluoride toothpaste | 208 (58.8%) |
| | Does not use fluoride toothpaste | 12 (3.4%) |
| | Parents unsure | 134 (37.9%) |
| Sugary snack consumption | At least once per day | 147 (41.5%) |
| | More than once per day | 203 (57.3%) |
| | Never | 4 (1.1%) |

[a]SOHPP: Students' oral health promotion program.

partially rejected the null inferential hypotheses, as among the sociodemographic factors, father's education level and government-employment of household heads were significantly associated with perspectives on the SOHPP.

Parents in this study demonstrated a generally positive overall appraisal of the SOHPP, with mean scores of 79% for awareness and 74% for satisfaction; additionally, 61.6% reported a perceived positive impact on their child's tooth-brushing behavior. These appraisal levels were comparable to those reported for other child oral health programs, such as Australia's *Bright Smiles Bright Futures* and Qatar's *Asnani* program [36,37]. In addition, the parents were most aware of and satisfied with the oral health education and fluoride therapy components, but, less so with the treatment-related components of oral health profiling and treatment need identification. This may reflect these components' less interactive and visible nature, and possibly unmet expectations if identified treatment needs were not followed-up and referred adequately, given the large student population the program covered and the complexity of organizing care for such population.

Fluoride therapy acceptance in the present study was high, with 76.6% of parents accepting the intervention for their children. This high acceptance rate contrasted with lower rates reported in a U.S. study (58.7%) [43], but partially aligned with studies in Laos, the UK, and the U.S. [44,45]. The parents in this study attributed most fluoride therapy refusals to logistical, procedural, or trust issues with school-based delivery, with safety concerns accounting for approximately one-third of their reasons. This contrasted with international findings, where fluoride therapy refusals were often due to fears of toxicity or developmental harm, low awareness, aesthetic concerns, personal choice, or media-fueled distrust [45–49]. This contrast may largely reflect differences in main study aims, as unlike the other cited studies, this study was dedicated to examining parental perceptions of a national child oral health promotion program. In this specific context, improving procedural clarity and communication will likely better improve fluoride therapy buy-in compared to focusing on safety or aesthetic concerns.

Although baseline data were not collected, parental reports of children's post-program oral health behaviors still provide valuable information. According to parents, fewer than two-fifths of children brushed their teeth twice daily after the SOHPP, while the rest brushed once a day (35.3%), less than once a day (19.5%), or not at all (6.5%), clearly falling short of the internationally recommended guideline [50]. This prevalence was lower than reported national twice-daily tooth-brushing averages for 6- and 12-year-olds (47.9% and 55.2%) [51], but considerably higher than the 9.3% prevalence of a prior study similarly conducted in Tehran [52]. Comparisons across these studies should be made cautiously, given their vast design, sampling, and measurement differences. Still, this comparison might indicate that although the SOHPP had a positive local impact, it fell short of national benchmarks. Interestingly, despite the suboptimal prevalence reported for twice-daily tooth-brushing, 61.6% of parents deemed that the SOHPP has positively impacted their child's

**Table 4. SOHPP[a] Awareness and Satisfaction by Socio-demographics among Parents of Tehran's Primary School Children (n = 354).**

| Variable | Category | Overall SOHPP Awareness | | | Overall SOHPP Satisfaction | | |
|---|---|---|---|---|---|---|---|
| | | Mean | SD | p | Mean | SD | p |
| Father's education level | Primary education or lower | 5.2 | 2.5 | < 0.001* | 5.1 | 2.2 | 0.011* |
| | Middle school education | 6.3 | 1.7 | | 5.9 | 1.7 | |
| | High school diploma | 6.5 | 1.7 | | 6.1 | 1.5 | |
| | Associate degree or higher | 6.6 | 2 | | 6 | 1.6 | |
| Mother's education level | Primary education or lower | 6 | 2.1 | 0.018* | 5.8 | 2 | 0.766 |
| | Middle school education | 6 | 2 | | 6 | 1.5 | |
| | High school diploma | 6.8 | 1.7 | | 6 | 1.7 | |
| | Associate degree or higher | 6.4 | 2 | | 5.8 | 1.7 | |
| Household head's occupation | Self-employed | 6.4 | 2 | 0.007* | 5.9 | 1.7 | 0.401 |
| | Private-sector laborer | 6.5 | 1.5 | | 6 | 1.9 | |
| | Private-sector employee | 6.4 | 2 | | 5.9 | 1.5 | |
| | Government laborer | 6.3 | 2.1 | | 6.1 | 1.9 | |
| | Government employee | 5.4 | 2.3 | | 5.4 | 1.6 | |
| | Unemployed | 4.6 | 2.2 | | 5.5 | 1.7 | |
| | Retiree | 7.3 | 0.8 | | 7 | 0.6 | |
| Household size | 3 | 6.4 | 1.9 | | 5.8 | 1.7 | |
| | 4 | 6.4 | 1.6 | | 6 | 1.8 | |
| | 5 | 6.3 | 2.3 | 0.196 | 5.9 | 1.5 | 0.985 |
| | 6 | 5.7 | 2.5 | | 5.9 | 1.7 | |
| | 7 or higher | 5.7 | 1.7 | | 5.7 | 1.7 | |
| Relation to the child | Mother | 6.4 | 2 | | 5.9 | 1.7 | |
| | Father | 6.4 | 1.8 | 0.12 | 5.9 | 1.7 | 0.719 |
| | Other | 5.7 | 2.4 | | 6 | 1.9 | |
| Child's school grade | First | 6.3 | 2 | | 5.7 | 1.8 | |
| | Second | 6.2 | 2 | | 5.9 | 1.8 | |
| | Third | 6.4 | 2 | 0.82 | 6 | 1.7 | 0.544 |
| | Fourth | 6.5 | 1.8 | | 6.1 | 1.5 | |
| | Fifth | 6 | 2.2 | | 5.6 | 1.8 | |
| | Sixth | 6.4 | 0.9 | | 5.6 | 2.3 | |

[a]SOHPP: Students' oral health promotion program; *P-values below 0.05 were deemed statistically significant.

tooth-brushing, highlighting a perception–behavior gap. Similarly, although only 3.4% of parents reported fluoride toothpaste non-use, a notable 37.9% were entirely unaware of their child's fluoride toothpaste usage. These findings suggest that parents' optimism, potentially stemming from their notable awareness and satisfaction with the SOHPP, did not align with actual post-program child oral health behaviors. As a result, greater emphasis on monitoring, reinforcement, and parental engagement seem to be necessary to enable efficient translation of perceived program benefits into actual behaviors. Regarding diet, about two-fifths of children reportedly consumed sugary snacks at least once daily, though the survey did not capture more frequent intake, likely underestimating true consumption. Actual intake is likely higher, with prior estimates suggesting about eight daily servings among Iranian children [53]. Thus, sugar exposure likely remained a behavioral risk post-SOHPP, despite the WHO's recommendations to limit sugary snacking [54]. Encouragingly, 83% of parents reported that the children mainly consumed healthy foods at school, suggesting the school environment reinforced

**Table 5. Sociodemographic Indicators of SOHPP[a] perceptions among parents of Tehran's primary school children (n = 354).**

| Overall awareness | | | |
|---|---|---|---|
| **Variable** | **B** | **CI** | **p** |
| Father's education level | 0.18 | 0.01 to 0.35 | 0.040* |
| Mother's education level | −0.01 | −0.163 to 0.15 | 0.919 |
| Household size | −0.20 | −0.407 to 0.01 | 0.062 |
| Household head's occupation | −1.16 | −1.950 to −0.37 | 0.004* |
| Relation to the child | −0.21 | −0.535 to 0.12 | 0.217 |
| Child's school grade | 0.03 | −0.133 to 0.18 | 0.747 |
| **Overall satisfaction** | | | |
| **Variable** | **B** | **CI** | **p** |
| Father's education level | 0.17 | 0.02 to 0.32 | 0.032* |
| Mother's education level | −0.08 | −0.22 to 0.05 | 0.235 |
| Household size | 0.000 | −0.19 to 0.19 | 1.000 |
| Household head's occupation | −0.67 | −1.38 to 0.04 | 0.063 |
| Relation to the child | −0.17 | −0.47 to 0.13 | 0.260 |
| Child's school grade | 0.05 | −0.09 to 2.00 | 0.466 |
| **Acceptance of fluoride therapy** | | | |
| **Variable** | **Exp (B)** | **CI** | **p** |
| Father's education level | 1.37 | 0.04 to 0.60 | 0.024* |
| Mother's education level | 0.88 | −0.37 to 0.11 | 0.276 |
| Household size | 0.85 | −0.47 to 0.14 | 0.293 |
| Household head's occupation | 0.40 | −1.89 to 0.07 | 0.068 |
| Relation to the child | 1.16 | −0.33 to 0.64 | 0.538 |
| Child's school grade | 1.12 | −0.12 to 0.34 | 0.359 |
| **Perceived impact on child tooth-brushing** | | | |
| **Variable** | **B** | **CI** | **p** |
| Father's education level | −0.06 | −0.25 to 0.12 | 0.511 |
| Mother's education level | 0.13 | −0.04 to 0.30 | 0.140 |
| Household size | −0.19 | −0.42 to 0.04 | 0.112 |
| Household head's occupation | −1.48 | −2.38 to −0.59 | 0.001* |
| Relation to the child | −0.17 | −0.53 to 0.19 | 0.360 |
| Child's school grade | 0.04 | −0.14 to 0.22 | 0.654 |

[a]SOHPP: Students' oral health promotion program *P-values below 0.05 were deemed statistically significant.

the program's nutritional efforts. Overall, while the SOHPP may have supported some positive oral health behaviors in its target population, notable gaps seem to persist, emphasizing the need for continued interventions.

Despite the often underappreciated role of fathers in child oral health, fathers' education emerged as a significant indicator of SOHPP perceptions in the present study, as higher-educated fathers demonstrated greater program awareness, higher program satisfaction, and stronger acceptance of fluoride therapy. This fits with evidence showing that higher education helps people find and use health information, is linked to higher socioeconomic status with better access to dental care and hygiene products, builds more trust in public health systems, and makes it easier to navigate healthcare services [55–60]. Interestingly, maternal education did not significantly indicate SOHPP perceptions, contrasting with the existing literature which generally identifies mothers as primary agents of child oral health and maternal education as a key indicator of related oral health behaviors [25,61]. Sociocultural patterns in Iran and the broader Middle East may explain

this discrepancy, as patriarchal norms in these regions often assign fathers primary authority in health decision-making, while mothers focus on daily caregiving tasks [62–64]. Since traditional assumptions about maternal influence seem to not apply in all cultural contexts, local family dynamics should be considered when implementing child oral health programs.

Household heads employed in the government sector were less likely to be aware of the SOHPP or to perceive a positive impact on their child's tooth brushing. Several factors may explain this. The structured, bureaucratic nature of government employment may have limited this demographic group's opportunities to encounter program information, since demanding careers have been shown to reduce parental engagement in child oral health matters [65,66]. Additionally, this group's awareness of public sector's inefficiencies may have fostered skepticism among them, as perceptions of corruption or ineffectiveness have been shown to lower trust in government-led programs [67]. Alternatively, these parents may already have higher oral health literacy and established tooth-brushing routines for their child, making the SOHPP seem less valuable to them, a notion partially supported by a study among French government employees [68]. These hypotheses warrant further investigation, particularly to guide targeted outreach for different subgroups of working parents.

Finally, although household size did not reach statistical significance for parents' SOHPP awareness or perceived impact on child tooth-brushing, the negative coefficients suggest a potential attenuation effect. This finding is noteworthy given that prior health-behavior studies have reported that larger households face competing demands on time and resources that might dilute the effectiveness of health messaging among them [69,70]. Future research should aim to more precisely explore the relationship between household composition and parental perceptions of child oral health initiatives.

This study has several strengths, including its comprehensive data collection, a relatively strong response rate of 67% for a telephone-based survey, the use of random sampling, a validated questionnaire, and a focus on parental perspectives, which represents a critical gap in the literature. The interviewer-administered, telephone-based format was selected to enhance response accuracy compared with self-administered surveys and to enable safe data collection during the COVID-19 pandemic. Several limitations should also be acknowledged. The cross-sectional design, though suitable for the research aim, restricted causal inference. Reliance on self-reports raised risks of recall and social desirability bias, likely amplified by interviewer-led calls, while phone-based recruitment may have excluded households without stable phone access. The TUMS jurisdiction was chosen for feasibility during COVID-19, but Tehran's greater socioeconomic resources and healthcare access compared to rural areas may have limited generalizability and inflated parental awareness, satisfaction, and child oral health behaviors. The absence of baseline data, consistent with the study's aim to focus on parental perspectives rather than clinical measures, reduced the ability to assess the SOHPP's impact. Additionally, the awareness and satisfaction indices were derived from a number of checklist-based items rather than broader latent constructs. This approach may have resulted in lower measurement granularity. However, it was a choice made intentionally to align with the study's objective of assessing awareness and satisfaction with specific program components, to accommodate an interviewer-administered telephone survey, and to minimize respondent burden. Finally, referring to all participants as "parents" for simplicity and consistency with the study aims, given that most (89%) identified as such, may have resulted in some overgeneralization.

Taking the above limitations into account, multi-level strategies can be proposed to strengthen parental engagement and buy-in to help maximize the long-term impact of Iran's SOHPP, and, with appropriate caution, of similar school-based programs in LMIC settings. The treatment-related components of the program, which received the lowest awareness and satisfaction ratings, would likely benefit from parent-friendly printed or online summaries of children's oral health profiles, accompanied by clear next steps, reminders, and follow-up mechanisms. High parental acceptance of fluoride therapy can serve as a strong foundation for expanding preventive care within the program, while refusals could be further reduced through clearer communication, advance notification, and greater procedural transparency. To address the gap between perceived program benefits and actual twice-daily tooth-brushing, reinforcement tools such as take-home tooth-brushing calendars or mobile game-based applications may be introduced. In parallel, healthier and less sugary snacking could be

promoted through parent-focused nutrition education sessions and partnerships with school food vendors. Parental socio-demographic characteristics may also be leveraged. For fathers with lower educational attainment, simplified program information delivered through videos, images, or voice messages may be effective, with more highly educated fathers engaged as advocates to promote this content. In addition, government-employee household heads may benefit from workplace-based outreach, such as targeted messaging via workplace communication platforms or informational posters displayed at worksites.

Future research could address this study's limitations through several strategies. Longitudinal studies tracking children's oral health behaviors and clinical outcomes before and after SOHPP implementation would enable causal inferences and quantify program impact. These studies should consider incorporating more objective measures of program impact, such as dental examinations or behavioral logs completed by children or teachers. Furthermore, including rural and semi-urban populations in sampling frames would enhance generalizability, and qualitative studies, such as focus groups or in-depth interviews with parents, would more deeply explore cultural and systemic barriers to SOHPP uptake. Employing more detailed psychometric assessments, such as multi-item Likert scales or factor analysis, can capture parental awareness and satisfaction with greater measurement precision. Additionally, the roles of household head occupation and household size in shaping parental perceptions of the program could be further explored.

## Conclusions

The present study provides a comprehensive assessment of parental perceptions of Iran's national child oral health program, the SOHPP, identifying both its strengths and areas for improvement. The findings suggest that further gains in parental awareness and satisfaction could be achieved by strengthening the program's treatment-related components through improved follow-ups and stronger referrals. Moreover, further reducing refusals of fluoride therapy will likely benefit from clearer communication and greater transparency. The observed gap between perceived program benefits and actual tooth-brushing behavior underscores the need for sustained post-program behavioral reinforcement, while continued daily sugary snacking points to the importance of school-based dietary guidance. Finally, outreach efforts should be tailored to key sociodemographic characteristics through strategies such as simplified briefings for fathers with lower education levels and workplace-based messaging for government-employee household heads. Collectively, these refinements could foster more equitable parental engagement, strengthen SOHPP implementation, and offer transferable insights for similar programs in other LMICs.

## Supporting information

**S1 File. English translation of the study questionnaire.**
(DOCX)

**S2 File. Anonymized dataset.**
(XLSX)

## Acknowledgments

We would like to wholeheartedly thank all those who contributed to this study, particularly Dr. Mohammad Javad Kharazi Fard for his assistance with statistical analysis. We would also like to acknowledge the use of GPT-5 (GPT-5, OpenAI, San Francisco, CA, USA) for editing this manuscript's text, with prompts to improve text conciseness, readability, and coherence.

## Author contributions

**Conceptualization:** Mohammad Reza Khami, Shabnam Varmazyari.

**Data curation:** Mohammadreza Naderi.



**Formal analysis:** Shabnam Varmazyari.

**Funding acquisition:** Mohammad Reza Khami.

**Investigation:** Mohammad Reza Khami, Mohammadreza Naderi.

**Methodology:** Mohammad Reza Khami, Mohammadreza Naderi, Shabnam Varmazyari.

**Project administration:** Mohammad Reza Khami, Shabnam Varmazyari.

**Resources:** Mohammad Reza Khami, Mohammadreza Naderi.

**Software:** Mohammadreza Naderi, Shabnam Varmazyari.

**Supervision:** Mohammad Reza Khami.

**Validation:** Mohammad Reza Khami, Shabnam Varmazyari.

**Visualization:** Mohammadreza Naderi, Shabnam Varmazyari.

**Writing – original draft:** Mohammad Reza Khami, Mohammadreza Naderi, Shabnam Varmazyari.

**Writing – review & editing:** Mohammad Reza Khami, Mohammadreza Naderi, Shabnam Varmazyari.

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
