## [Decision Letter · Decision Letter 0]

28 Nov 2025

Dear Dr. Varmazyari,

Thank you for submitting your manuscript to PLOS ONE. After having commented on your submission (please see our comments given below), and after careful consideration, we feel that it has merit but does not fully meet PLOS ONE’s publication criteria as it currently stands. Therefore, we invite you to submit a revised version of the manuscript that addresses the points raised during the review process.

We look forward to receiving your revised manuscript.

Kind regards,

Andrej M Kielbassa

Academic Editor

PLOS ONE

2. In the ethics statement in the Methods, you have specified that verbal consent was obtained. Please provide additional details regarding how this consent was documented and witnessed, and state whether this was approved by the IRB.

“This study was supported in part by funding from Tehran University of Medical Sciences as a student thesis grant.”

4. We are unable to open your Supporting Information file [S2 File.sav]. Please kindly revise as necessary and re-upload.

Reviewers' comments:

Reviewer's Responses to Questions

**Comments to the Author**

1. Is the manuscript technically sound, and do the data support the conclusions?

Reviewer #1: No

Reviewer #2: Yes

2. Has the statistical analysis been performed appropriately and rigorously?

Reviewer #1: Yes

Reviewer #2: Yes

3. Have the authors made all data underlying the findings in their manuscript fully available?

Reviewer #1: No

Reviewer #2: Yes

4. Is the manuscript presented in an intelligible fashion and written in standard English?

Reviewer #1: Yes

Reviewer #2: Yes

Reviewer #1: This submitted draft would seem interesting, is considered easily intelligible, and might be worth following after some revisions and clarifications.

Title

- Please add type of study.

Abstract

- Please stick to Journal style.

- Please revise for uniform Journal style, see "p = 0.040".

- When it comes to your conclusions, please exclusively stick to your aims. Remember that you wanted To "explore parents’ perceptions of Iran’s SOHPP, the sociodemographic factors shaping them, and children’s post-program oral health behaviors". Do not simply repeat your results here. Do not speculate. Do not provide well-accepted (but meaningless) phrases. Instead, provide a reasonable and generalizable extension of your outcome.

Intro

- More than two full pages would not be adequate for such a study, Please simply elaborate both aims and objectives, and present a sound rationale. A one-page Intro would seem adequate.

- Please adapt your reference style, and use square brackets. "(...) of school-aged children worldwide (1, 2)." must read "(...) of school-aged children worldwide [1, 2]." Revise thoroughly.

- At the end of this section, please remember that you have statistically analyzed your data, so, consequently, a sound null hypothesis must be presented. The latter must be deducible from the foregoing thoughts.

Meths

- Please remember that Plos One will consider publishing qualitative research only if it adheres to appropriate study design and reporting guidelines, as described in the submission guidelines. For example, "qualitative data sources include, but are not limited to, interviews, (...), and free-form answers to questionnaires and surveys". For more information, please go to https://journals.plos.org/plosone/s/submission-guidelines#loc-qualitative-research. You have submitted the outcome of "phone-surveyed parents of primary school children", and such a study might be semi-quantitative, or even qualitative. Please clarify.

- Additionally, please go to https://journals.sagepub.com/doi/10.1177/1049732315617444, and discuss your set-up with reference to your sample size.

- With this section your define your study as being "cross-sectional". You surely will know that cross-sectional studies are a type of observational studies. The latter must be pre-registered, please see https://jamanetwork.com/journals/jamanetworkopen/fullarticle/2836842. Please provide you a-priori registration date and number.

- Your sample size calculation would not seem clear. Please provide more details. See comments given above.

- With your calculated sample size being 347, why did you approach "525 individuals"? Please clarify.

- Same with " Four centers, Ayat, Farmafarmanian, Imam Hassan Mojtaba, and Avicenna, were randomly selected from this list." Please clarify your randomization approach.

- Note that with ALL materials and methodologies (including statistical software), please use general/non-proprietary names with your text, followed by (brand name; manufacturer, city, ST[ate - abbreviated, if US], country) in parentheses. Stick to semicolon. Revise thoroughly.

Results

- Again, compare "p = 0.040", "p = .011", "p = 0.040" and "P-value". Revise for uniform Journal style. Consulting some recently published Plos One papers should be helpful.

Disc

- Stick to H0 when starting this section.

- Do not provide a literature review here. This section will benefit from a sound DISCUSSION.

Concl

- This section would not seem satisfying. Remember that "Conclusions" does not mean "Summary" (or "Repetition of Outcome"). Revise carefully.

- Again, with your Conclusions, please stick exclusively to your aims. Do not simply repeat your results here. Do not speculate. Do not provide well-accepted (but meaningless) phrases. Instead, provide a reasonable and generalizable extension of your outcome.

Refs

- Your references list must be adapted to Journal style. Remember that this is considered your task, to avoid any mistakes.

In total, several shortcomings and drawbacks do not allow for any proceeding with this draft, and re-review is considered mandatory.

Reviewer #2: The study employed a cross-sectional analytical design, which is well suited to exploring parents’ perceptions, awareness, and satisfaction with the Students’ Oral Health Promotion Program (SOHPP). Sampling was carried out across four randomly selected comprehensive healthcare centers, yielding a 67% response rate and a total of 354 participants—an adequate and reasonably representative sample. The questionnaire was carefully developed and underwent simplified validation through assessments of face and content validity, along with test–retest reliability, which is appropriate considering the descriptive focus of the measures. Statistical analyses, including ANOVA, chi-square tests, and regression models with backward elimination, were appropriate for the study objectives. The authors verified assumptions such as multicollinearity and model fit, which were found to be acceptable. Ethical approval was obtained from the relevant institutional review board, and informed consent procedures were clearly described and properly implemented.

The study’s main findings—high levels of parental awareness and satisfaction, notable sociodemographic differences (particularly related to fathers’ education and occupation), and the presence of a perception–behavior gap—are clearly supported by both the descriptive and inferential data. The interpretation of results is balanced, recognizing the strengths of the program as well as its limitations, including the cross-sectional design, reliance on self-reported information, and limited generalizability. The conclusions are consistent with the evidence presented and appropriately emphasize the need to strengthen treatment-related elements, improve communication, and adapt outreach efforts to different parental backgrounds.

Comments to the Authors

The introduction could be strengthened by more clearly outlining why parents’ perspectives are critical for the long-term success and sustainability of school-based oral health programs. It would also help to clarify how this study addresses existing gaps in the regional or international literature.

Regarding questionnaire validation, the manuscript mentions face and content validity and a test–retest approach. Including additional information—such as the time interval between tests or any calculated reliability values—would make the instrument’s credibility clearer.

Because the awareness and satisfaction indices were based on only a few items, it might be worth acknowledging this as a methodological limitation. You could suggest that future studies use a more detailed psychometric assessment, such as reliability testing or factor analysis, to improve measurement precision.

The statistical approach is well chosen and explained. However, the use of backward regression could be better justified. A brief explanation of why this method was selected, and whether sensitivity analyses were done to confirm model stability, would enhance transparency.

The discussion is well developed and connects effectively with previous research. Still, the paper would benefit from a stronger emphasis on practical implications—particularly how the observed links between fathers’ education, government employment, and program perceptions could guide the design of future oral health interventions or communication strategies.

**Do you want your identity to be public for this peer review?** For information about this choice, including consent withdrawal, please see our Privacy Policy

Reviewer #1: No

Reviewer #2: No

---

## [Author Response · Author response to Decision Letter 1]

23 Jan 2026

The response to reviewers file is attached.

---

## [Decision Letter · Decision Letter 1]

11 Feb 2026

Parents’ perspectives on a national child oral health promotion program: sociodemographic influences and behavioral insights – a cross-sectional analysis

PONE-D-25-51769R1

Dear Dr. Varmazyari,

We’re pleased to inform you that your manuscript has been judged scientifically suitable for publication and will be formally accepted for publication once it meets all outstanding technical requirements.

Kind regards,

Prof. Dr. Dr. h. c.Andrej M Kielbassa

Academic Editor

PLOS One

Additional Editor Comments (optional):

Reviewers' comments:

Reviewer's Responses to Questions

**Comments to the Author**

Reviewer #1: All comments have been addressed

Reviewer #2: All comments have been addressed

2. Is the manuscript technically sound, and do the data support the conclusions?

Reviewer #1: Yes

Reviewer #2: Yes

3. Has the statistical analysis been performed appropriately and rigorously?

Reviewer #1: Yes

Reviewer #2: Yes

4. Have the authors made all data underlying the findings in their manuscript fully available?

Reviewer #1: Yes

Reviewer #2: Yes

5. Is the manuscript presented in an intelligible fashion and written in standard English?

Reviewer #1: Yes

Reviewer #2: Yes

Reviewer #1: With the help of the reviewers, this revised and re-submitted draft has been considerably improved. Still, there would seem minor typos, and, unfortunately, the references again have not been adapted to uniform Journal style. These shortcomings might be handled together with the typesetter.

Reviewer #2: The revised manuscript satisfactorily addresses the points raised in the initial review. The study design is now clearly stated, the abstract and references follow journal style, and the introduction has been streamlined with the addition of an explicit null hypothesis. The authors also clarify the quantitative nature of the study, provide clearer explanations of the sample size calculation, recruitment strategy, and random selection of centers, and ensure consistent reporting of statistical results.

Overall, the manuscript is clearer and more transparent. Methodological details, data availability, and ethical procedures are adequately documented, the discussion is more focused on interpretation, and the conclusions are aligned with the study aims rather than repeating results. I confirm that the issues raised previously have been addressed in the revised version, which now meets the criteria for publication in PLOS ONE.

**Do you want your identity to be public for this peer review?** For information about this choice, including consent withdrawal, please see our Privacy Policy

Reviewer #1: No

Reviewer #2: No

---

## [Editor Report · Acceptance letter]

PONE-D-25-51769R1

PLOS One

Dear Dr. Varmazyari,

I'm pleased to inform you that your manuscript has been deemed suitable for publication in PLOS One. Congratulations! Your manuscript is now being handed over to our production team.

Kind regards,

on behalf of

Prof. Dr. med. dent. Dr. h. c. Andrej M Kielbassa

Academic Editor

PLOS One